# Influences of age and gender on operative risks following carotid endarterectomy: A systematic review and meta-analysis

Sothida Nantakool[1&], Busaba Chuatrakoon[2&], Saritphat Orrapin[3], Rachel Leung[4], Dominic P. J. Howard[4], Amaraporn Rerkasem[1], José G. B. Derraik[1,5,6,7], Kittipan Rerkasem[1,8,9]*

**1** Research Institute for Health Sciences, Environmental—Occupational Health Sciences and Non Communicable Diseases Research Group (EOHS and NCD Research Group), Chiang Mai University, Chiang Mai, Thailand, **2** Faculty of Associated Medical Sciences, Department of Physical Therapy, Chiang Mai University, Chiang Mai, Thailand, **3** Faculty of Medicine, Department of Surgery, Division of Vascular Surgery, Thammasat University (Rangsit Campus), Pathum Thani, Thailand, **4** Nuffield Department of Clinical Neurosciences, Centre for Prevention of Stroke and Dementia, University of Oxford, Oxford, United Kingdom, **5** Liggins Institute, University of Auckland, Auckland, New Zealand, **6** Department of Women's and Children's Health, Uppsala University, Uppsala, Sweden, **7** Faculty of Medical and Health Sciences, Department of Paediatrics, Child and Youth Health, University of Auckland, Auckland, New Zealand, **8** Faculty of Medicine, Department of Surgery, Chiang Mai University, Chiang Mai, Thailand, **9** Faculty of Medicine, Clinical Surgical Research Center, Chiang Mai University, Chiang Mai, Thailand

&These authors contributed equally to this work.
* rerkase@gmail.com

**Data Availability Statement:** All relevant data are within the paper and its Supporting Information files.

## Abstract

### Objectives

This review aims to undertake a comprehensive review of the literature and investigate associations of age and gender on 30 days post carotid endarterectomy (CEA) and up to 5 years post CEA stroke, death, and combined stroke and death.

### Design

A systematic review and meta-analysis.

### Methods

Three main electronic databases including the Cochrane Library, MEDLINE, and Embase were searched from their inception to July 2022. Studies examining operative risks (i.e., stroke, death, and combined stroke and death following CEA) linked to age or gender were included. Two independent reviewers were responsible for study selection, quality assessment, and data extraction. Odds ratio (OR) and 95% confidence interval (CI) of all outcomes were calculated.

### Results

44609 studies were retrieved from the search. There were 127 eligible studies (80 studies of age, 72 studies of gender, 25 studies of age and gender) for pooling in the meta-analysis.

**Funding:** The author(s) received no specific funding for this work.

**Competing interests:** The authors have declared that no competing interests exist.

With regards to stroke and death risks within 30 days post CEA; patients aged ≥75 had higher death (OR 1.38; 95% CI 1.10–1.75) than patients aged <75. Patients aged ≥80 had higher stroke risk (OR 1.17; 95% CI 1.07–1.27) and death risk (OR 1.85; 95% CI 1.48–2.30) particular in asymptomatic patients (OR 2.4; 95% CI 1.56–3.81). Pooled effect estimates by gender, at 30 days post CEA, showed that female was associated with increased risk of stroke (OR 1.28; 95% CI 1.16–1.40), with more risk in asymptomatic female patients (OR 1.51; 95% CI 1.14–1.99).

## Conclusions

This meta-analysis highlights that older people is associated with increased stroke risk, particularly asymptomatic octogenarians who had higher likelihood of death within 30 days post CEA. In addition, female especially those with asymptomatic carotid stenosis had greater likelihood of stroke within 30 days post CEA surgery.

## Introduction

The effectiveness of carotid endarterectomy (CEA) for prevention of ipsilateral ischaemic stroke has been established in randomised clinical trials and meta-analysis. Trials often exclude older patients (above 75 years old) which constitute a significant portion of both asymptomatic and symptomatic cases receiving CEA in clinical practice nowadays (40% patients in German registry and 43.3% in NVR versus 30.4% in meta-analysis of 4 RCTs) [1]. It remains uncertain whether these trial-based observation is generalisable to older populations. Elderly patient with symptomatic carotid stenosis on best medical treatment have higher risk of recurrent stroke and may yield greater benefit from CEA [2]. Although routine carotid endarterectomy is not recommended for asymptomatic carotid stenosis, patients with over 70% degree of stenosis and high perceived risk of stroke by imaging markers may potentially benefit from CEA [3]. The association between age and the short- and long-term safety and efficacy of CEA may be better studied with up-to-date evidence including all study types.

Gender is accounted as a factor contributing to stroke and death risks following CEA. A past meta-analysis including all studies of symptomatic and asymptomatic stenosis have showed higher risk of postoperative adverse events in females [4]. In addition, symptomatic stenosis trials showed higher perioperative stroke and death in female and past guidelines had suggested that females may benefit less from CEA [5–8]. However, accumulating evidence from more recent studies showing the conflicted results with the finding of the previous meta-analysis [7,9–11]. Such trials have suggested that female was associated with decreased stroke risks following CEA. Taken together, our objective is to undertake a comprehensive review of the literature and investigate associations of age and gender on 30 days post CEA (short-term) and up to 5 years post CEA (long-term) stroke, death, and combined stroke and death.

## Materials and methods

### Search strategy

The meta-analysis was performed based on the Preferred Reporting Items for Systematic reviews and Meta-Analyses (PRISMA) guidelines [12]. Three electronic databases including the Cochrane Library [1993- July 2022], MEDLINE [1966-July 2022], and EMBASE [1980-July 2022]. To obtain comprehensive searches, hand searching was also carried out by checking

reference lists from the identified articles. The search term used to identify relevant studies was "carotid endarterectomy" (S1 Table). This meta-analysis was registered in the PROSPERO database (CRD42021276031).

## Study selection

Two independent reviewers (BC, SN) were responsible for title and abstract screening. Eligible studies were screened based on the following criteria: (i) studies investigating 30 days post CEA) and/ or up to 5 years post CEA risk differences (i.e., a combined stroke and death, stroke, and death in people who underwent CEA) linked to age or gender, (ii) containing participants age >18 years old, and (iii) no language restriction. Any study carrying out a carotid stenting, conducting a combined CEA with carotid stenting without a single data of CEA, or containing participants who had bilateral simultaneous carotid endarterectomy without reporting data separately for participants who underwent a unilateral procedure were excluded. Where there was disagreement of eligible studies, this was resolved by a discussion between the two reviewers or reviewed by the third reviewer (KR).

## Risk of bias and quality assessment

Risk of bias was measured using Risk of Bias in Non-randomized Studies-of Interventions (ROBINS-I) tool for observational studies [13]. This tool rates eligible studies into five categories: low risk, moderate risk, serious risk, critical risk, and no information. The Grading of Recommendations Assessment, Development and Evaluation (GRADE) approach was carried out to evaluate the overall quality of eligible evidence [14]. The quality of eligible evidence was assessed on the basis of risk of bias as well as inconsistency, indirectness, imprecision, and publication bias of the data. The GRADE is interpreted as high, moderate, low, or very low quality. Two independent reviewers (BC and SO) were responsible for these assessments. A third reviewer (KR) was involved in case of a dispute between the two reviewers.

## Data extraction

Two reviewers (SN, BC) extracted publication details (author, publication year, study design, and study country), characteristics of participants in each study (number of participants, mean age, gender, surgical details), and information on operative risks (stroke, death, or both). The corresponding author was sent an email to obtain additional information in case of incomplete data identified. If no response was received within a week, the result was excluded. Any disagreement was resolved by a consensus of the two reviewers.

## Statistical analyses

All outcome measures were analysed using an event number and total number of participants in exposure group (i.e., older people or females) and control group (i.e., younger people or males). A pooled meta-analysis was presented as an odds ratio (OR) and 95% confidence interval (CI). A minimum two included studies were considered to pool in meta-analysis. We separated and performed meta-analysis in studies reporting short-term (30 days post-surgery) and long-term outcome (i.e., 4 years and 5 years). A random effects model, using Mantel-Haenszel method was employed as the main method for all analyses. If there were moderate heterogeneity across studies, based on $I^2$ 30%-60%, together with p value of Cochrane's Q <0.1 [15], we have attempted to seek possible factors using subgroups analysis. The subgroup analyses were determined by characteristics of carotid stenosis symptoms (i.e., symptomatic, asymptomatic, and mixed), while the sensitivity analyses were conducted by removing studies affecting the

influence plot and any studies that could not clearly identify recruitment periods. We evaluated publication bias, measured by funnel plot and Begg's test when there was sufficient number of included studies (≥10 included studies). All analyses were performed using STATA Statistical Software, version 14.2 (StataCorp LP, USA).

## Results

A total of 44609 studies were obtained from the three electronic databases and reference lists. After duplicates were removed, 43884 studies were identified for title screening. Of these, 3398 studies were recorded for abstracts screening. Of the 155 studies remaining for full text screening, 22 studies were excluded due to no relevant outcomes and 6 studies were removed due to identifying same databases and timeframe. Finally, 127 publications (S2 Table and S1 File) were eligible for qualitative and quantitative analyses. Of 127 studies, there were 80 studies of age, 72 studies of gender, and 25 studies of age and gender (Fig 1).

### Effect of age on stroke, death, and combined stroke and death risks following carotid endarterectomy

**Short-term effects (30 days post-surgery).**   Separated meta-analyses of cut off ages of 75 and 80 were reported. Pooled effect estimates found that patients with age ≥75 had a 38% significantly higher risk of death (Fig 2) and a 22% significantly greater risk of combined stroke and death (S1.1 Fig in S1 Fig) than younger. Another pooled effect estimate showed that patients with age ≥80 had a 17% significantly higher stroke (Fig 3), an 85% significantly higher risk of death (S1.2 Fig in S1 Fig), and a 33% significantly greater combined stroke and death risk (S1.3 Fig in S1 Fig) than younger. No difference of a stroke risk between patients with age ≥75 vs <75 was observed (S1.4 Fig in S1 Fig). Regarding subgroup analysis by carotid stenosis symptoms, only cut off age of 80 was analysed. Of this, asymptomatic patients aged ≥80 had a 144% significantly increased death risk than asymptomatic patients aged <80, while a symptomatic subgroup showed similar effect between groups (S4 Table). Sensitivity analysis was also performed in the cut off age of 80. After removing studies which might affect the influence plot [16,17], the result showed an 87% significantly increased risk of death in patients aged ≥80. We further considered removed studies containing unclear recruitment period [18,19] and found an 82% significantly higher risk of death in patients aged ≥80 (S5 Table). As such analyses, effect estimates still be showed a trend of increased risk of death following CEA among older patients.

### Effect of gender on stroke, death, and combined stroke and death following carotid endarterectomy

**Short-term effects (30 days post-surgery).**   The result showed that female was significantly associated with a 28% increased risk of stroke (Fig 4). In line with stroke risk, female had a 21% significantly increased risk of combined stroke and death (S1.5 Fig in S1 Fig). There was no association between gender and risk of death (S1.6 Fig in S1 Fig) Subgroup analysis was stratified by carotid stenosis symptoms. Among symptomatic patients, we found that female was significantly correlated with a 48% increased risk of stroke and a 26% increased risk of combined stroke and death. Correlations between female with significant increases in such operative risks were more enhanced among asymptomatic patients (stroke risk: OR 1.51, 95%CI 1.14–1.99; combined stroke and death: OR 1.50, 95%CI 1.17–1.92) (S4 Table). A sensitivity analysis was then performed after omitting studies affecting the influence plot [6,17]. After removal, female was associated with a 26% significantly increased stroke risk and a 21% significantly increased combined stroke and death risk (S5 Table). We considered to remove

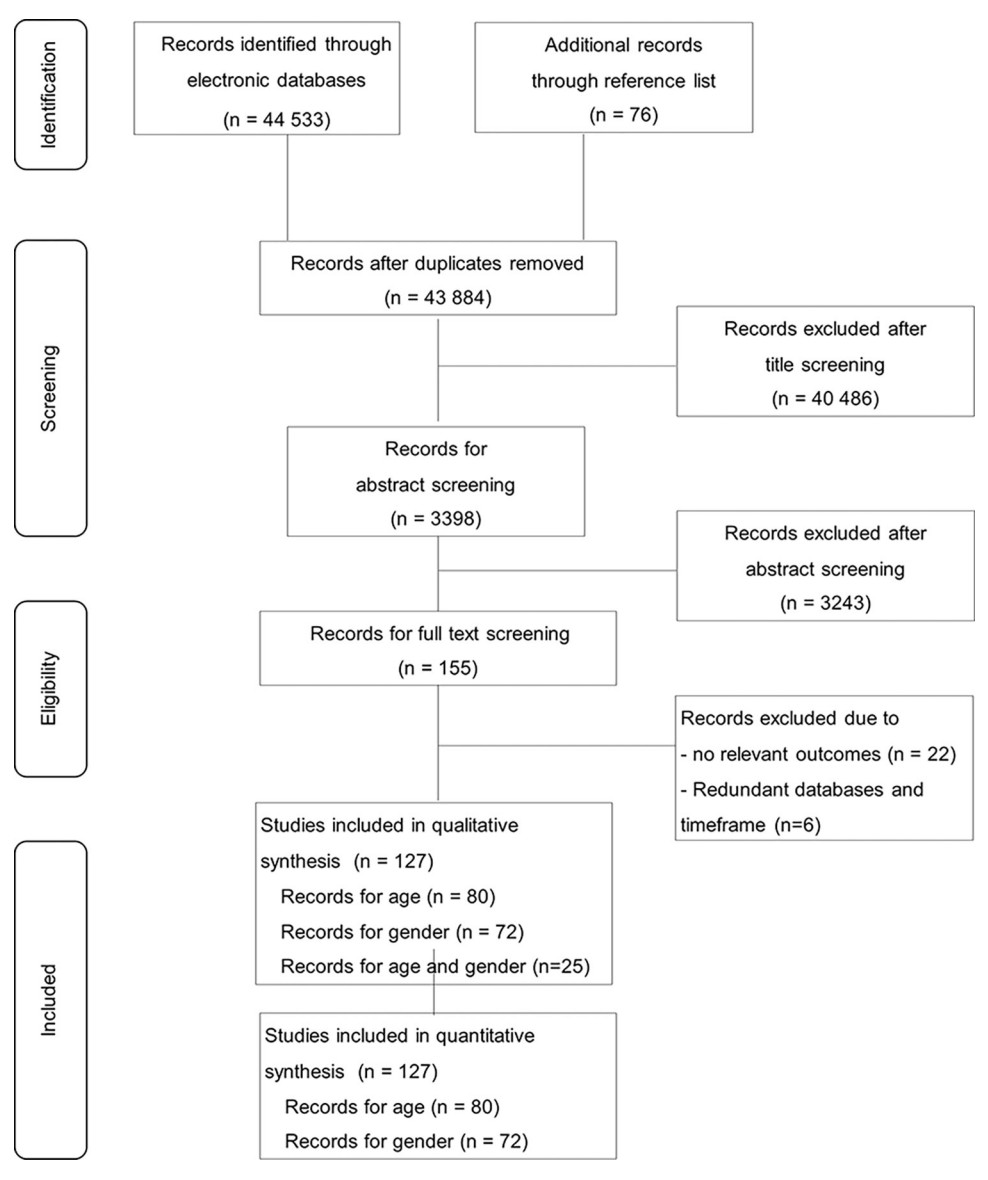

**Fig 1. PRISMA flow chart.**

studies that had unclear recruitment periods [20–26]. After excluding, female was associated with a 29% significantly increased risk of stroke and a 24% significantly increased risk of combined stroke and death (S5 Table). Such analyses found that the effect estimate still be revealed trends toward increased risk of operative risks in female.

**Long-term effects (up to 5 years post-surgery).** There were 2 separated meta-analyses including 5 years stroke risk and 4 years combined stroke and death risk following CEA between genders. Of these, no differences of such operative risks between female and male were demonstrated (S1.7-S1.8 Figs in S1 Fig, respectively).

## Publication bias of included studies

Publication bias assessment performed on studies reporting outcomes at 30 days post-surgery showed no apparent publication bias (S2.1–2.9 Figs in S2 Fig). Except for stroke, death, and

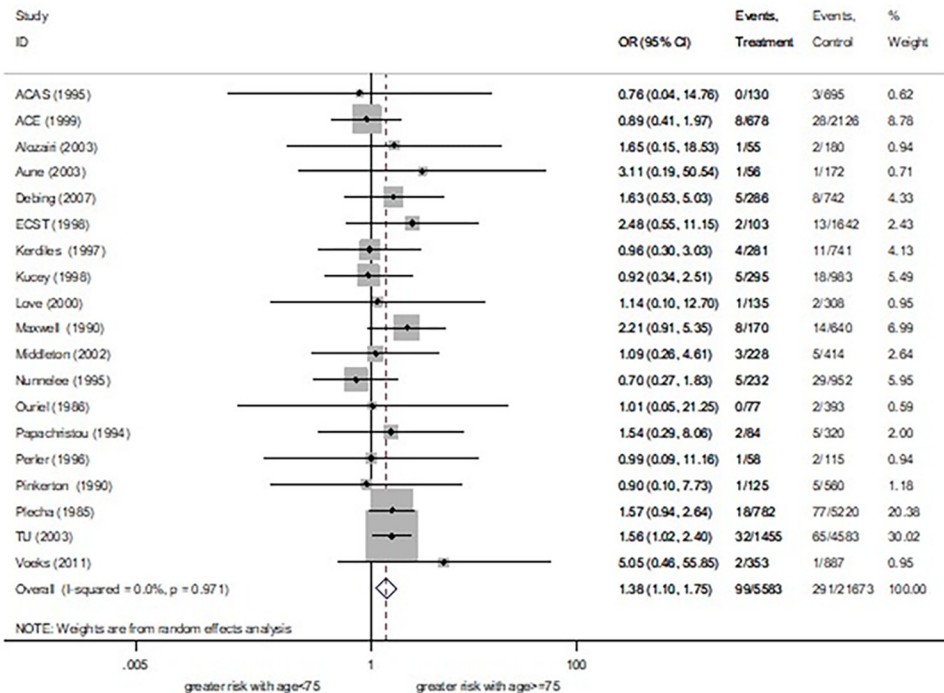

**Fig 2. 30 days death risk between age ≥75 years vs <75 years.**

combined stroke and death risks at cut off age 80, and stroke risk between female and male. After performing trim-and-fill method, pooled effect estimates of the determined outcomes were 2%, 23%, 2%, and 21% changes from the former pooled effect estimates (S2.2, 2.4, 2.6 and 2.7 Figs in S2 Fig). We could not determine the publication bias for included studies investigating operative risks during long-term follow up due to insufficient number of included studies.

## Risk of bias and quality assessment

Most included studies had no information (60%), followed by serious (22%), low (16%), and moderate (2%) risk of bias (S3 Table). Eligible evidence reporting 30 days post-surgery outcomes by age were of low- to very low-quality. Eligible evidence reporting 30 days post-surgery, 4 and 5 years follow-up outcomes by gender were also of very low-quality (Table 1).

## Discussion

In this comprehensive meta-analysis of all study types of age and gender, we confirmed an overall association between age and postoperative stroke, death, and combined stroke and death. Second, asymptomatic octogenarian patients had an obvious large effect on higher likelihood of death within 30 days post CEA. Third, females had higher risk of stroke and combined stroke and death, particularly asymptomatic female patients had more risks following 30 days post CEA.

The findings of significantly greater postoperative risks following CEA in older individuals is consistent with a previous meta-analysis [4]. On the other hand, a recent meta-analysis with individual patient data of up to 4000 patients from four symptomatic stenosis trials showed that age was not associated with higher risk postoperative stroke and/or death [1]. Our findings refuted such findings and confirmed a higher postoperative stroke and/or death in older

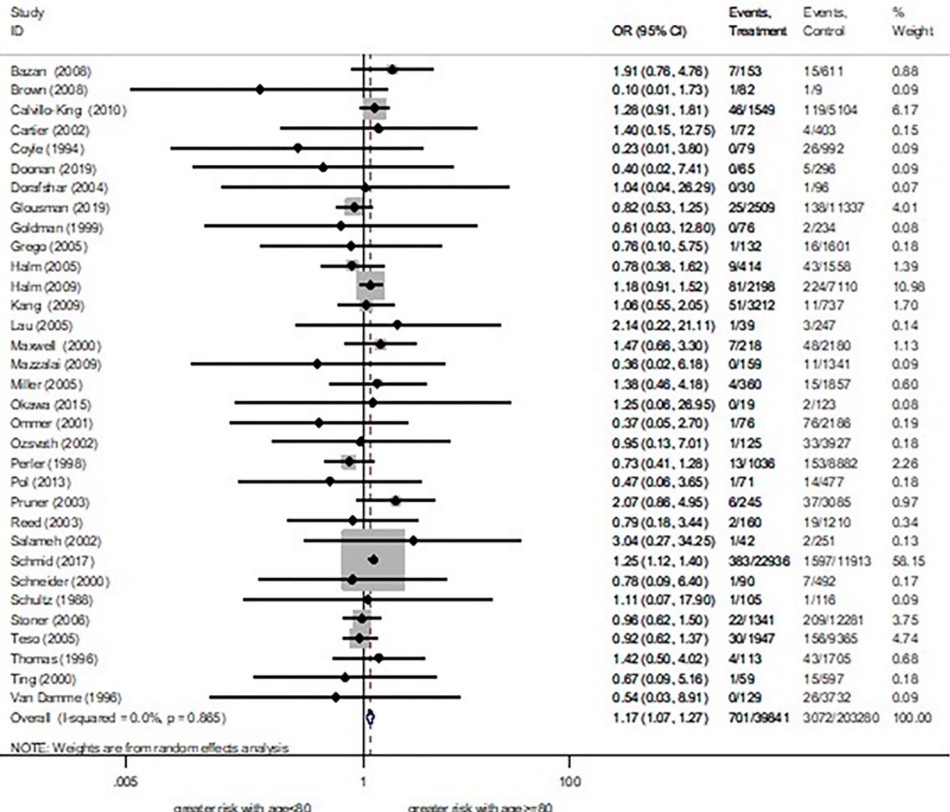

**Fig 3. 30 days stroke risk between age ≥80 years vs <80 years.**

patients, and especially amongst octogenarian. This may reflect those results on age related outcomes from trials are subjected to patient selection bias. In fact, the prevalence of cardiovascular risk factors in octogenarians including hypertension (78%), diabetes (20%), dyslipidaemia (59%), smoking (6%) and prior history of cardiovascular diseases (59%) may be dissimilar to the prevalence of cardiovascular risk factors of our included studies [1].

Our finding that higher likelihood of mortality in older people, particularly asymptomatic octogenarians as compared to their counterparts is supported by previous evidence reporting that advanced age is associated with an elevated risk of mortality in CEA, especially apparent in octogenarians [18]. A single included study has discussed that small hospital, inadequate resources, and few volumes of surgeon and/ or few experience surgeons are likely to influence higher rate of mortality in older people compared to younger people [27]. This finding may be partially explained by the fact that, older people undergoing CEA usually had more comorbidities compared to their younger counterparts, which may contribute to a higher fatality rate [28]. Thus, a cut-off age of ≥80 years with asymptomatic type is likely to be more concern when this aged group is being considered for CEA.

Interestingly, the current meta-analysis revealed that less benefit of CEA in a prevention of a perioperative stroke risk in females particularly with asymptomatic type than males may partially be explained by the gender and symptomatic type differences in a carotid plaque phenotype [29]. More fibrous plaque volume, and unstable plaque, which have been evident as a predictive of stroke are commonly found in males than females [29]. Several large trials have documented that, after CEA, high baseline risk of stroke in males is attenuated to an equal

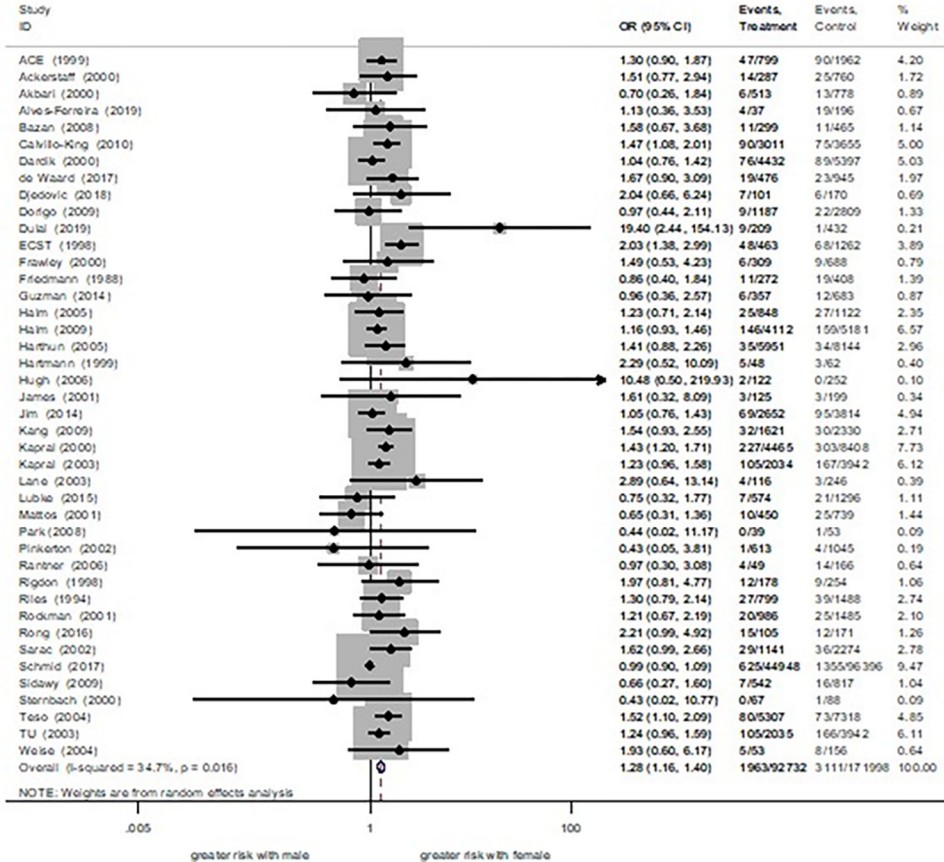

**Fig 4. 30 days stroke risk between female vs male.**

level in females [5,6,30]. It is likely that unstable plaque phenotypes more common in males benefit more from CEA than the stable plaque phenotype more commonly seen females. Another possible factor is the degree of vulnerability of plaques. More presence of embolization at pre-operative CEA period in females associates with a developing in higher number of post-operative embolization when compared to males [31]. Higher number of post-operative CEA embolization in females, compared to males has further been contributed to a risk of cerebrovascular complication [32]. Regarding the gender-associated differences, the symptom is also accounted for difference plaque features [29]. Comparing between symptomatic and asymptomatic types, some morphologic features of plaque are changed when it becomes symptomatic type [29]. Previous evidence has documented that plaque rupture, fibrous cap thinning, infiltration of the fibrous cap with foam cells, and intraplaque fibrin are identified as unstable plaque, which is more commonly found in people with symptomatic stenosis [33]. Thus, females who have asymptomatic stenosis had very high stable plaque, leading to have less benefit from CEA.

## Strengths and limitations

There are some strengths that should be highlighted. First, we have attempted to reduce the risks of potential biases through a comprehensive and exhaustive search of both published and unpublished evidence, with no language restrictions, and contacts with authors in case of incomplete data. In addition, the raw data underwent a rigorous review process by

**Table 1. Summary of findings.**

| Outcomes | Illustrative comparative risks* (95% CI) | | Relative effect (95%CI) | Number of participants (studies) | Quality of evidence (GRADE) | Comments |
|---|---|---|---|---|---|---|
| **Risk factor of age** | **Assumed risk (<75 yr)** | **Corresponding risk (≥75 yr)** | | | | |
| **Stroke (30 days post-surgery)** | 36 per 1,000 | 39 per 1,000 (33 to 47) | OR 1.09 (0.91,1.31) | 24,016 (18) | □ΘΘΘ Very low[a,b] | - |
| **Death (30 days post-surgery)** | 13 per 1,000 | 18 per 1,000 (14 to 23) | OR 1.38 (1.10,1.75) | 27,256 (19) | □ΘΘΘ Very low[a,b] | - |
| **Combined stroke-death (30 days post-surgery)** | 45 per 1,000 | 55 per 1,000 (48 to 63) | OR 1.22 (1.07,1.39) | 29,629 (25) | □ΘΘΘ Very low[a,b] | - |
| | **Assumed risk (<80 yr)** | **Corresponding risk (≥80 yr)** | **Relative effect (95%CI)** | **Number of participants (studies** | **Quality of evidence (GRADE)** | **Comments** |
| **Stroke (30 days post-surgery)** | 15 per 1,000 | 18 per 1,000 (16 to 19) | OR 1.17 (1.07,1.27) | 243,121 (33) | □ΘΘΘ Very low[a,b] | - |
| **Death (30 days post-surgery)** | 7 per 1,000 | 13 per 1,000 (11 to 16) | OR 1.85 (1.48,2.30) | 233,397 (33) | □ΘΘΘ Very low[a,b] | 3 studies have no event in both groups for analysis |
| **Combined stroke-death (30 days post-surgery)** | 20 per 1,000 | 27 per 1,000 (24 to 30) | OR 1.33 (1.21,1.48) | 227,367 (32) | □□ΘΘ Low[a] | - |
| **Risk factor of gender** | **Assumed risk (male)** | **Corresponding risk (female)** | **Relative effect (95%CI)** | **Number of participants (studies)** | **Quality of evidence (GRADE)** | **Comments** |
| **Stroke (30 days post-surgery)** | 18 per 1,000 | 23 per 1,000 (21 to 25) | OR 1.28 (1.16,1.40) | 264,730 (42) | □ΘΘΘ Very low[a,b] | - |
| **Death (30 days post-surgery)** | 9 per 1,000 | 9 per 1,000 (8 to 10) | OR 0.96 (0.84,1.09) | 171,387 (32) | □ΘΘΘ Very low[a,b] | One study has no event in both groups for analysis |
| **Stroke and death (30 days post-surgery)** | 25 per 1,000 | 30 per 1,000 (39 to 43) | OR 1.21 (1.09,1.34) | 164,024 (46) | □ΘΘΘ Very low[a,b] | - |
| **Stroke (5 years post-surgery)** | 85 per 1,000 | 111 per 1,000 (111 to 292) | OR 1.30 (0.49,3.44) | 1,833 (2) | □ΘΘΘ Very low[a,c] | - |
| **Combined stroke-death (4 years post-surgery)** | 35 per 1,000 | 40 per 1,000 (15 to 108) | OR 1.15 (0.43,3.09) | 915 (2) | □ΘΘΘ Very low[a,c] | - |

CI, confidence interval; GRADE, Grading of Recommendations, Assessment, Development and Evaluations.

[a] Downgrade 2 levels due to observational study.

[b] Downgrade 1 level due to serious risk of bias.

[c] Downgrade 1 level due to wide range of CI.

independent reviewers who involved in study selection, risk of bias and quality assessment, and data extraction.

Despite the several strengths identified, this review has some limitations that should be concern. First, the associations between age and gender with operative risks following CEA were

derived on the basis of very low quality of evidence. The two important factors contributing to the reduced evidence quality are observational study design and serious risk of bias. The serious risk of bias is attributed to dissimilar baseline characteristics (i.e., hypertension, diabetes mellitus, and smoking history) [34–37] between groups. Second, the evidence quality is likely to be underestimated due to the limitations in risk of bias interpretation. Among the included studies, most of them was rated as no information on the risk of bias (60%) which limited us to rate for the quality of evidence. The major reason is that there was no separated data on the baseline characteristics between older and younger people, and between females and males. Perhaps age and gender were not the primary determinant of interest in most included studies. Third, pooled effect estimates of this meta-analysis with unadjusted confounders are accounted as a significant limitation. As age and gender among most identified studies were not the main determinant of interest, we could not extract the adjusted effects. Thus, the meta-analysis finding should be interpreted with caution. Last, it is unclear whether patients recruited into the studies received best medical therapy for prevention of ischaemic stroke. Current guidelines recommend CEA for asymptomatic stenosis patients with over 70% stenosis and a perceived high risk of stroke based on imaging markers [3]. Unfortunately, our meta-analysis was not able to capture age-related outcomes with degree of stenosis and imaging results to assess whether CEA performed in included studies were appropriate. We also were not able to ascertain if patients in our included studies were on best medical therapy for risk factor control. Nonetheless, the association between older age and higher risk of death and combine stroke and death may warren a careful selection of older patients with asymptomatic stenosis to be considered for a prophylactic carotid intervention.

## Conclusions

This meta-analysis highlights that older people is associated with an increased stroke risk, particularly asymptomatic octogenarians who had higher likelihood of death within 30 days post CEA surgery. In addition, female people especially those with asymptomatic carotid stenosis had greater likelihood of stroke within 30 days post CEA surgery.

## Supporting information

**S1 Checklist. PRISMA 2020 checklist.**
(DOCX)

**S1 Fig. Other main results.**
(PDF)

**S2 Fig. Publication bias.**
(PDF)

**S1 Table. Search strategy.**
(PDF)

**S2 Table. Characteristics of included studies.**
(PDF)

**S3 Table. Risk of bias.**
(PDF)

**S4 Table. Subgroup analyses of 30 days stroke, death, and combined stroke death risks of age and gender.**
(PDF)

**S5 Table. Sensitivity analyses of 30 days stroke, death, and combined stroke death risks of age and gender.**
(PDF)

**S1 File. References of included studies.**
(PDF)

## Acknowledgments

This study was partially supported by Chiang Mai University.

## Author Contributions

**Conceptualization:** Sothida Nantakool, Busaba Chuatrakoon, Saritphat Orrapin, Amaraporn Rerkasem, Kittipan Rerkasem.

**Formal analysis:** Busaba Chuatrakoon.

**Methodology:** Sothida Nantakool, Busaba Chuatrakoon, Saritphat Orrapin, Rachel Leung, Dominic P. J. Howard, Amaraporn Rerkasem, José G. B. Derraik.

**Writing – original draft:** Sothida Nantakool, Busaba Chuatrakoon.

**Writing – review & editing:** Sothida Nantakool, Saritphat Orrapin, Rachel Leung, Dominic P. J. Howard, José G. B. Derraik.

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
