## [Decision Letter · Decision Letter 0]

4 Jan 2023

PONE-D-22-30639Influences of age and gender on operative risks following carotid endarterectomy: a systematic review and meta-analysisPLOS ONE

Dear Dr. Rerkasem,

Thank you for submitting your manuscript to PLOS ONE. After careful consideration, we feel that it has merit but does not fully meet PLOS ONE’s publication criteria as it currently stands. Therefore, we invite you to submit a revised version of the manuscript that addresses the points raised during the review process.

We look forward to receiving your revised manuscript.

Kind regards,

Athanasios Saratzis

Academic Editor

PLOS ONE

Journal Requirements:

2. We note that this manuscript is a systematic review or meta-analysis; our author guidelines therefore require that you use PRISMA guidance to help improve reporting quality of this type of study. Please upload copies of the completed PRISMA checklist as Supporting Information with a file name “PRISMA checklist”.

Additional Editor Comments:

The 3 expert Reviewers have identified some major issues relating to the methodology as well as the presentation of the article. Please consider all points raised by the Reviewers, especially relating to the way the analyses were conducted and clinical translation/impact of variables that might have influenced the results.

Reviewers' comments:

Reviewer's Responses to Questions

**Comments to the Author**

1. Is the manuscript technically sound, and do the data support the conclusions?

Reviewer #1: Partly

Reviewer #2: Yes

Reviewer #3: No

2. Has the statistical analysis been performed appropriately and rigorously? 

Reviewer #1: No

Reviewer #2: Yes

Reviewer #3: No

3. Have the authors made all data underlying the findings in their manuscript fully available?

Reviewer #1: Yes

Reviewer #2: Yes

Reviewer #3: Yes

4. Is the manuscript presented in an intelligible fashion and written in standard English?

Reviewer #1: Yes

Reviewer #2: No

Reviewer #3: No

5. Review Comments to the Author

Reviewer #1: Many thanks for the opportunity to review this manuscript reporting the results of a meta-analysis of studies reporting age and/or sex associations with outcomes following carotid endarterectomy. This is a large and apparently rigorous review of available literature which has included and pooled data from a large number of studies. Whilst this offers breadth, my main concern with the methodology of this meta-analysis is heterogeneity, particularly in the meta-analysis of associations of age with outcomes. Whilst I commend the authors on their work, I think major revisions are required to the analyses before I could recommend this manuscript for publication. Please find detailed comments below:

1. It is very rare that fixed-effect analyses are suitable. I do not think fixed-effects analyses are appropriate in this type of meta-analysis. The authors have used a mix of observational and randomised data and as such the heterogeneity of study designs is too great for fixed-effects analyses to be used. It is also not clear from the reporting which meta-analyses the authors have performed have used a fixed-effects model. The authors should repeat any analyses where a fixed-effects model has been used and instead use a random effects model.

2. Whilst I accept that the use of I^2 as a measure of heterogeneity is widely used in the literature, it is frequently used and interpreted inappropriately (https://bmcmedresmethodol.biomedcentral.com/articles/10.1186/1471-2288-8-79). It definitely should not be used to decide on whether a fixed- or random-effects analysis should be used. As above, please use a random-effects model for all analyses. Consider reporting Tau^2 in addition to I^2 as this represents the ammount of between-study heterogeneity whilst I^2 estimates the % of variability due to between-study heterogeneity and precision of the effect estimates has a big impact on I^2.

3. I do not think the meta-analyses of age in its current form is valid. At the very least the forest plot should be presented in subgroups of studies that compared ages at different cut points (<65 and 65+; <70 and 70+; <75 and 75+; <80 and 80+). However I'm not sure that pooling all these studies with different cut points is appropriate as this pools data from different age cohorts. The most robust way to investigate the influence of age would be to perform separate meta-analyses pooling data from studies that have used the same cut-point only.

4. Similarly I have concerns about the meta-analyses of short- and long-term outcomes. The time points for these outcomes are not defined in the methods. Given that the outcomes are reported as odds ratios then the outcome measures themselves need to be measured at the same timepoint (e.g. 30 day mortality pooled together, 1 year mortality pooled together, 5 year mortality pooled together). Studies reporting outcome measures at different timepoints should not be pooled in the same analysis.

5. There is a notable lack of table summarising included studies presented in the manuscript or supplementary material. The authors state all the data that has been extracted but this is not presented anywhere. I think there should be a table included which describes the data included. Of particular note, there is no information of sample size for each individual study presented anywhere in either the manuscript or supplementary material. A table presenting the raw data extracted from each study as detailed in the methodology and also including the sample size should be presented in the supplementary material. This would provide important context to enable readers to interpret the results appropriately.

6. Are the authors confident that there is no double-counting of participants? For example, have more than one study utilised data from the same registry e.g. ACS-NSQIP, VSQIP, NVR etc. The authors need to take care not to include studies that have utilised the same database with overlapping timeframes in the same meta-analysis as this risks potentially including the same participants and outcome events in the same meta-analysis multiple times. This risk is highlighted in this recent paper by Hussein et al (https://bmcpublichealth.biomedcentral.com/articles/10.1186/s12889-022-14213-6).

7. I presume all of the data extracted and pooled are unadjusted for confounders. If so, this is a significant limitation to the study and should be mentioned. There is a very high risk that any observed effect is due to confounding in the individual studies and that pooling data magnifies this effect. It would be interesting, if possible, to include a meta-analysis of suitable outcome measures using effect estimates from studies that used comparible adjutment models to reduce the impact of confounding. This would also have limitations (especially as it is unlikely adjustment models were the same between studies) but may increase the confidence of findings.

8. I am unclear from the methods and reporting of results what sensitivity analyses have been performed. Subgroup analyses are usually presented as a single forest plot divided into relevant subgroups. Sensitivity analyses are reporting of the pooled analyses with only studies with specific characteristics removed from the analyses. Please could the authors be more specific about the subgroup analyses performed and what sensitivity analyses have been performed with their justification?

Minor comments:

1. The objectives in the abstract merely describe the methodology. Please describe the purpose of the meta-analysis based on the PICO question(s) i.e. investigate associations of age and sex on outcomes following CEA.

2. Please include the dates when the searches were last completed. They are mentioned in the abstract but not included in the manuscript that I can see (apologies if I've missed it).

3. I'm not sure what this means: "Hand searching which was proceeded via contacting expertise to further obtain published and unpublished studies." (lines 109-110) Please revise.

4. The order of reporting results seems strange. Whilst reporting risk of bias before presenting the results of the meta-analyses is usual, reporting the risk of publication bias and the GRADE assessments before presenting the actual results of the meta-analyses themselves seems strange. I think the more appropriate order would be to present the publication bias results after the results from the meta-analyses and then the GRADE assessments as the final part of the results.

Reviewer #2: This study aimed to stratify by age and gender, the 30-day and 5-year stroke and mortality outcomes following carotid endarterectomy. The authors undertook an extensive search of the literature including 133 studies. They concluded that individuals over the age of 80 and those with asymptomatic disease had a higher mortality rate within 30 days of carotid endarterectomy. Additionally, women had a higher incidence of stroke within 30 days of carotid endarterectomy than men. While this study is of interesting and beneficial value, the article is, at times, difficult to read and to interpret with grammatical errors. Below are detailed reviews of each section and include suggestions to improve the manuscript.

ABSTRACT:

Objective to determine the outcomes of CEA however the authors only reported the stroke and mortality rates. The word outcomes suggest that other factors were included such as TIA and MI. Please reword this throughout the paper to make it clear to the reader that stroke and mortality were the only outcomes assessed.

Line 47: First time CEA is mentioned, should be written out

INTRODUCTION:

Line 82: Please provide a reference for this statement

The second paragraph reads like a discussion proving an argument rather than setting the scene for the research undertaken. Consider rewording and restructuring.

Objective – Please provide a definition for what is meant by short-term and long-term (i.e. 30 days post CEA and up to 5 years). This should also be clearly defined in the methods. The first time the reader reads this is in the results.

METHODS:

Line 140: Please explain what was meant by ‘Incomplete data was clarified by contacting an author of the study’. Was there a standard protocol followed? i.e. An email was sent to the author and if no response was heard within 2 weeks the results were not included.

Statistical analysis: A random effect model should be used throughout rather than a fixed effect model.

RESULTS

Results section is difficult to read. If Odds Ratio and CI are reported in a table which is referenced then don’t include in the main body of the text as it is hard to follow. The effect of age and effect of gender are particularly difficult paragraphs to read, please restructure.

Line 165: Split the studies into; age, gender, age and gender. This will make it easier for the reader

Line 186: Where does this number come from? First time referenced.

Table 2: Should define what their presenting symptoms were and carry out subgroup analyses accordingly as there are important differences in risk between different symptomatic indications. E.g. categorising symptomatic patients as; stroke, ocular event, cerebral TIA. Carrying out analysis by symptomatic or asymptomatic is too vague. This should be reflected in the discussion.

DISCUSSION

Line 250: It is unclear where these percentages come from, is this from your data?

Line 269: Reference

Line 300: Reference the current Guideline

Reviewer #3: A priori women and older people do worse in elective vascular surgery.

Although gender may be associated with worse outcomes, are there any clinical confounding variables that may contribute to this?

Please ask an English speaker e.g. Dr Howard to correct the English in the document please. It is substandard for a good scientific journal.

Line 82 please reference ESVS carotid guideline

Endpoints: one of the clinical concerns long term, is whether women with smaller arteries have a higher restenosis rate. Can the authors include this outcome please?

It is interesting to see the peri-operative stroke risks are similar. This is reassuring in terms of efficacy of surgery.

I think the odds ratios may be scientifically interesting but are too low (OR<2) to really influence clinical practice.

Older people with asymptomatic disease are not offered surgery – the ACST finding that older people have higher risks is the explanation, therefore this finding is redundant.

It is no surprise that older people have more strokes and die more frequently than younger people. This finding does not require a meta-analysis.

There are several studies demonstrating the lower risk of stroke or death follow treatment of asymptomatic patients – e.g. ACST1 and 2, in comparison to symptomatic patients. Please comment.

I do not think that you can say that the risks are higher in asymptomatic females v symptomatic females as the 95%CI of the OR actually overlap. Please comment.

Similar comment for older patients. Please comment.

Rothwell’s subgroup paper suggested that older symptomatic patients benefit more from CEA – would the author’s like to comment - i.e. procedural risks may be higher, but non-operated stroke risks are also higher.

6. PLOS authors have the option to publish the peer review history of their article (what does this mean?). If published, this will include your full peer review and any attached files.

Reviewer #1: **Yes: **John Houghton

Reviewer #2: No

Reviewer #3: **Yes: **I do not see the clinical need for this study. The overlapping OR do not appear to show any separation between older and female asymptomatic v symptomatic patients.

---

## [Author Response · Author response to Decision Letter 0]

31 Mar 2023

Division of Vascular and Endovascular Surgery

 Department of Surgery, Faculty of Medicine 

Chiang Mai University, Chiang Mai, Thailand, 50200 

Tel +66 (0)81 8846638 

Email: rerkase@gmail.com

March 31, 2023

RE: submission of manuscript title: Influences of age and sex on operative risks following carotid endarterectomy: a systematic review and meta-analysis 

Dear Editor

I would like to thank for providing me an opportunity to revise the manuscript. We have responded all comments from the Reviewers. It is very useful for improving my manuscript. Hopefully, this revised version of the manuscript would meet the criteria of your journal to be published. 

If you need any further information, please do not hesitate to contact me. 

Yours sincerely 

Kittipan Rerkasem 

Professor Kittipan Rerkasem

---

## [Editor Report · Decision Letter 1]

26 Apr 2023

Influences of age and gender on operative risks following carotid endarterectomy: a systematic review and meta-analysis

PONE-D-22-30639R1

Dear Dr. Rerkasem,

We’re pleased to inform you that your manuscript has been judged scientifically suitable for publication and will be formally accepted for publication once it meets all outstanding technical requirements.

Kind regards,

Athanasios Saratzis

Academic Editor

PLOS ONE
---

## [Editor Report · Acceptance letter]

2 May 2023

PONE-D-22-30639R1 

Influences of age and gender on operative risks following carotid endarterectomy: a systematic review and meta-analysis 

Dear Dr. Rerkasem:

I'm pleased to inform you that your manuscript has been deemed suitable for publication in PLOS ONE. Congratulations! Your manuscript is now with our production department. 

Kind regards, 

on behalf of

Dr. Athanasios Saratzis 

Academic Editor

PLOS ONE